# Statistical Analysis of Uniform Switching Characteristics of Ta_2_O_5_-Based Memristors by Embedding In-Situ Grown 2D-MoS_2_ Buffer Layers

**DOI:** 10.3390/ma14216275

**Published:** 2021-10-21

**Authors:** Soeun Jin, Jung-Dae Kwon, Yonghun Kim

**Affiliations:** 1Department of Advanced Materials Engineering, University of Science and Technology (UST), 217 Gajeong-ro, Yuseong-gu, Daejeon 34113, Korea; zx95921@kims.re.kr; 2Department of Energy and Electronic Materials, Surface Materials Division, Korea Institute of Materials Science (KIMS), 797 Changwondaero, Sungsan-gu, Changwon 51508, Korea

**Keywords:** memristor, conductive bridge random-access memory (CBRAM), molybdenum disulfide (MoS_2_), non-volatile resistive memory, AP-PECVD

## Abstract

A memristor based on emerging resistive random-access memory (RRAM) is a promising candidate for use as a next-generation neuromorphic computing device which overcomes the von Neumann bottleneck. Meanwhile, due to their unique properties, including atomically thin layers and surface smoothness, two-dimensional (2D) materials are being widely studied for implementation in the development of new information-processing electronic devices. However, inherent drawbacks concerning operational uniformities, such as device-to-device variability, device yield, and reliability, are huge challenges in the realization of concrete memristor hardware devices. In this study, we fabricated Ta_2_O_5_-based memristor devices, where a 2D-MoS_2_ buffer layer was directly inserted between the Ta_2_O_5_ switching layer and the Ag metal electrode to improve uniform switching characteristics in terms of switching voltage, the distribution of resistance states, endurance, and retention. A 2D-MoS_2_ layered buffer film with a 5 nm thickness was directly grown on the Ta_2_O_5_ switching layer by the atomic-pressure plasma-enhanced chemical vapor deposition (AP-PECVD) method, which is highly uniform and provided a superior yield of 2D-MoS_2_ film. It was observed that the switching operation was dramatically stabilized via the introduction of the 2D-MoS_2_ buffer layer compared to a pristine device without the buffer layer. It was assumed that the difference in mobility and reduction rates between Ta_2_O_5_ and MoS_2_ caused the narrow localization of ion migration, inducing the formation of more stable conduction filament. In addition, an excellent yield of 98% was confirmed while showing cell-to-cell operation uniformity, and the extrinsic and intrinsic variabilities in operating the device were highly uniform. Thus, the introduction of a MoS_2_ buffer layer could improve highly reliable memristor device switching operation.

## 1. Introduction

In the past few decades, information processing technology regarding mobile edge devices has been developed in accordance with to the rapid population of the Big Data and Internet of Things (IoT) era, whereby the amount of data has improved explosively [1]. However, conventional von Neumann architecture has a physically separated structure of memory and computing data; with with frequent data transfer process, the transfer bottleneck of delay with increasing power consumption dramatically increases. Therefore, it is necessary to newly design an efficient computing architecture that simultaneously possesses a massively parallel structure and extreme low energy consumption [2,3]. Extensive efforts have been conducted to realize an energy-efficient parallel computing architecture called a neuromorphic system, which would provide great potential applications, such as parallel data processing and unstructured pattern recognition [4]. Various types of non-volatile memory devices have been proposed for use in neuromorphic core hardware, such as resistive random-access memory [5,6,7] phase-change RAM [8,9] spin-transfer torque magnetoresistive RAM (STT-MRAM) [10,11], and conventional flash memory [12], respectively. Among them, RRAM devices are promising candidates for use in overcoming structural problems of von Neumann computing, because with these devices, computing and storage functions can be performed in the same circuit, enabling processing-in-memory (PIM) computing [13,14,15,16,17]. RRAM has a simple structure of metal–insulator–metal (MIM) configuration; thus, it provides a high-density, fast switching speed, low power consumption, and good compatibility with the conventional CMOS architecture [18,19]. However, RRAM devices need to improve their inherent variability due to the random movement of ions for extrinsic (device-to-device) and intrinsic (cycle-to-cycle) aspects [20].

Meanwhile, two-dimensional (2D) layered materials such as graphene, MoS_2_, and WS2 were recently applied in electronic or optoelectronic devices because they have unique physical properties with ultrathin thicknesses and mechanical flexibility [21,22]. Among various 2D materials, molybdenum disulfide (MoS_2_) has a low power consumption, band gap tunability, high effective mobility, and rapid switching speed, which show that MoS_2_ is a promising candidate for use in vertical or lateral electrical devices such as field-effect transistors (FETs), non-volatile memory devices, junction diodes, and flexible optical sensors [23,24,25]. More recently, various two-terminal memristor devices have been reported which use two-dimensional MoS_2_ materials, such as ultrathin [26,27], flexible [28], and photonic memristor. Furthermore, a multilayered heterosynaptic device using 2D-NbSe_2_ and HfO_2_ has demonstrated the ability to control conductive metal filaments. As a result, the stochastic switching behavior was effectively suppressed and improved in synaptic function [29]. However, 2D-NbSe_2_ film should be transferred onto a HfO_2_ switching layer, inevitably inducing non-ideality factors such as the limitation of large-area uniformity, wrinkle defects, and residue of the PMMA supporting layer. Thus, the technology of high production yield with large-area and uniform growth onto high-k switching film is required.

In this study, a buffer layer of 2D-MoS_2_ film was introduced in Ta_2_O_5_-based RRAM between the Ta_2_O_5_ and Ag electrode, where 2D-MoS_2_ film was sulfurized from molybdenum trioxide (MoO_3_) using the low-temperature AP-PECVD method. The crystal structure and chemical composition of the MoS_2_ film were investigated. By introducing a MoS_2_ buffer layer, the random resistive switching behavior was dramatically stabilized. The experimental data of MoS_2_ buffered device show the ohmic conduction mechanism in all devices. Additionally, the exceptional device yield was achieved in 98% of 100 device cells.

## 2. Results and Discussion

Figure 1a shows the scheme of the fabricated memristor device structure; the device had a 250 nm via-hole structure comprising a Pt bottom electrode (BE), a switching layer of Ta_2_O_5_ (10 nm), a buffer layer of 2D-MoS_2_ (5 nm), and a Ag top electrode (TE) of 70 nm. To measure the electrical properties, the bottom electrode (BE) was grounded and external voltage was applied to the top electrode (TE). To investigate the existence and quality of the AP-PECVD-synthesized 2D-MoS_2_ film, the physical and chemical properties were inspected. Figure 1b shows the cross-sectional image of the 2D-MoS_2_ crystalline structure obtained via transmission electron microscopy (TEM), and transmission electron microscopy with energy-dispersive X-ray spectroscopy (TEM-EDS) mapping was carried out to verify the configuration of MoS_2_, including the Mo and S elements. Meanwhile, the Si element in SiO_2_ used as a substrate was also observed. It was confirmed that the layered two-dimensional crystal structure was successfully synthesized, resulting in the number of layers being three or four. Figure 1c shows the Raman spectrum of the 2D-MoS_2_ film, and the main two peaks were observed due to the inherent vibrational modes of the Mo and S atom. The main two peaks of 379 cm^−1^ (E^1^_2g_) and 403 cm^−1^ (A_1g_) indicated the in-plane and out-of-plane vibrational mode of the Mo and S atoms. In addition, to analyze the chemical bonding of 2D-MoS_2_, X-ray photoelectron spectroscopy (XPS) measurement was conducted, as shown in Figure 1d,e. A doublet for Mo 3d_3/2_ (232.7 eV), Mo 3d_5/2_ (229.4 eV), and S 2s (226.7 eV) represent Mo^4+^ components of 2H-MoS_2_. Additionally, doublet peaks of S 2p_3/2_ (162.4 eV) and S 2p_1/2_ (163.6 eV) corresponding to 2H-MoS_2_ were also found.

To investigate effect of the 2D-MoS_2_ buffer layer on the atomic switching characteristics, DC I–V switching curves of Ta_2_O_5_-based devices depending on the presence of the 2D-MoS_2_ thin 2D layer are shown in Figure 2a,b. Figure 2a shows the typical I–V curve of the pristine Ta_2_O_5_-based switching device without any 2D-MoS_2_ buffer layer. In this operation, the compliance current (I_CC_) for the set process was limited to 0.5 mA to prevent permanent breakdown. Additionally, a positive voltage of 1.5 V was applied for the set process (from high resistive state (HRS) to low resistive state (LRS)), and the negative voltage of −1 V was applied for the reset process (from LRS to HRS). The pristine Ta_2_O_5_ device initially showed a typical non-volatile switching characteristic. However, the switching characteristics deteriorated within five cycles. In particular, the reset operation of the negative voltage region almost disappeared, which resembles the unipolar switching characteristics. In contrast, the Ta_2_O_5_-based switching device with the 2D-MoS_2_ buffer layer showed a highly stable non-volatile switching characteristic. In this operation, the set voltage was 1.5 V, the reset voltage was −0.5 V, and the I_CC_ was 0.5 mA. Interestingly, the switching characteristics were highly stable within 20 cycles. To investigate the statistical distribution of the two different devices, the cumulative distribution function (CDF) was shown as a function of two resistance states with LRS and HRS. The read voltage was measured at 0.05 V after the set and reset processes. In the pristine Ta_2_O_5_-based switching device, it displayed a very unstable switching operation, which showed the transition fail region at CDF 40%. In contrast, the Ta_2_O_5_/MoS_2_ RS device indicated a more stable non-volatile operation without any degradation and it had a sufficient on/off ratio of about 350. These results indicate that the device operation reliability could improve considerably by introducing a 2D-MoS_2_ buffer layer, which could effectively control the mobility and redox rate of the Ag atom.

To investigate the current conduction mechanism of the Ta_2_O_5_-based device with a 2D- MoS_2_ buffer layer, the slope values for the set and reset operations were extracted from a double logarithmic plot of I–V curves, as shown in Figure 3a,c. For the set process, as shown in Figure 3a, the resistance states changed very sharply, meaning that by fitting the switching curves, the slope values were extracted around 1. Additionally, the slope values for the reset process were obtained as nearly 1. It could be elucidated that the conduction mechanism for 2D-MoS_2_-buffered devices follows the ohmic conduction mechanism, which is due to thermally produced free charge carriers. When positive bias was applied, the Ag was oxidized and Ag ions migrated toward BE due to the external electrical field, and the reduction reaction occurred at BE. Thus, the Ag conduction filament was formed, and the device converted into LRS [30].
*Ag**→**Ag^+^ + e^−^* (TE)
*Ag^+^ + e^−^**→**Ag* (BE)

In the pristine Ta_2_O_5_ device, the electric field across the Ta_2_O_5_ layer could form weak and scattered Ag filaments. Subsequently, when the external voltage was turned off, the Ag filament contracted due to a random volatile phenomenon. Otherwise, in the Ta_2_O_5_ device with the 2D-MoS_2_ buffer layer, the atomically thin 2D-MoS_2_ film could effectively localize the Ag ions due to due to the combination effects of the layered structure and localized defect sites, as shown in Figure 3b. The plausible ion conduction mechanism via the insertion of a 2D-MoS_2_ buffer layer can be explained as follows.

Firstly, the Ag filament in the 2D-MoS_2_ buffer layer was formed with a cone shape, which is due to the difference in mobility and reduction rate between Ta_2_O_5_ and MoS_2_, resulting in a blocking effect of ion migration that occurred at the interface. Then, charges were concentrated in a portion of the dielectric layer, forming a much more stable filament in the Ta_2_O_5_ layer. In contrast, when negative voltage was applied, the filament was dissolved, and the device converted to a HRS. Then, the filament started to break from the relatively weak part. As a result, it broke from the filament at the interface between MoS_2_ and Ta_2_O_5_, as shown Figure 3d.

On the other hand, to further investigate the switching properties of reliable non-volatile memory devices, retention and endurance characteristics were measured, as shown Figure 3e,f. In retention measurement, various temperature environments at 50, 75, and 100 °C were applied, and it was measured by applying constant read voltages of 0.05 V. In all temperature environments, LRS and HRS keep a large enough memory margin for up to 10^4^ s. In the endurance test, the voltage was switched under DC, which was the same voltage condition for the set and reset operations shown in Figure 2b. It was confirmed that the device could operate over 800 cycles by retaining a sufficient on/off margin. These results indicate that the introduction of 2D buffer layers greatly improves the stability in the memory device applications.

To identify cell-to-cell uniformity, a total of 100 cells (within 10 by 10 matrix) were selected in a 20 by 20 matrix, which is the area highlighted by a red solid line in Figure 4a. Figure 4c,d indicates the set and reset current mapping images, respectively, and black colored squares represent an electrical shorted cell. In that area, an excellent device yield of 98% is shown, and the defect rate could be lowered if a cleaner depositional environment was introduced. Except for two shorted cells, all of them worked well. The operable cells also had a sufficient on/off ratio of about 270, as shown Figure 4b.

Statistical analysis was conducted to examine uniformity parameters related to operating conditions. The parameters from cell-to-cell and cycle-to-cycle voltage conditions were extracted from DC I–V curves. In addition, the statistical histograms were fitted to Gaussian function to extract the mean (μ), standard deviations (σ), and the coefficient of variation (σ/μ), which was multiplied by 100 and expressed as a percentage. Figure 5a shows the forming voltage range obtained from 50 device cells. Additionally, μ and σ/μ values are estimated to be 4.9 and 1.35%, respectively. Figure 5b shows the set voltage range, and the μ of voltage by cycle and cell is about 0.75 V for both, and the σ/μ is 16% and 39%, respectively. Cell-to-cell variation is lager but all of the device cells could operate within 1.5 V. Figure 5c shows the reset voltage range, and the μ and σ/μ of voltage by cycle and cell is −0.15, 18%, and −0.35, 15%, and where μ is negative, σ/μ is the absolute value. The cause for the variation is the filament branches; these grow competitively during repetitive switching operation. However, in the reset process, the μ is lower at cycle deviation, and this implies that an operation scheme becomes more uniform during repetitive operation.

## 3. Conclusions

In this study, 2D-MoS_2_ buffer layer was directly inserted in a Ta_2_O_5_-based memristor via an in situ AP-PECVD process in order to improve non-volatile memory characteristics. A MoS_2_ buffer layer was successfully synthesized on a Ta_2_O_5_ switching layer using the AP-PECVD method to obtain superior production yield, and the highly uniform and superior yield of MoS_2_ film was confirmed in a previous study. Additionally, we confirm the physical and chemical properties of MoS_2_ film, and it shows an excellent synthetic state. In the Pt/Ta_2_O_5_/Ag device, the volatile and non-volatile RSs occur randomly. However, the Pt/Ta_2_O_5_/MoS_2_/Ag device shows a highly reliable switching performance with minimized device failure and an on/off ratio of 350. It should be noted that the MoS_2_ buffer layer induces the bottleneck of Ag+ ion in the Ta_2_O_5_ and MoS_2_ interface, and it helps to form more substantial conduction filament. In addition, an excellent yield of 98% could be obtained with good operational uniformity, and extrinsic and intrinsic variability in device operation were highly uniform. Thus, the introduction of a MoS_2_ buffer layer could aid in improving the non-volatile characteristics of reliable memory devices.

## 4. Experimental Section

Firstly, a Ti adhesion layer was deposited on a 250 nm via-hole patterned silicon wafer, and then Pt BE was deposited, and for both, sputtering was used. Subsequently, a SiO_2_ insulation layer was deposited via plasma-enhanced CVD on 250 nm via-hole patterned by photolithography and reactive-ion etching processes. Next, Ta_2_O_5_ (10 nm) was deposited using plasma-enhanced atomic deposition, and a reactive chamber and TaF_5_ precursor were heated at 200 °C and 70 °C, respectively. Additionally, the stainless steel feeding line was heated at 100 °C to prevent TaF_5_ condensation. The Ta_2_O_5_ deposition cycle is as follows: TaF_5_ precursor was injected for 5 s while 60 sccm Ar gas was flowed to convey TaF_5_; a 10 s purge was carried out using Ar gas; H_2_O reactor was injected for 0.3 s; a 10 s was carried out purge. Then, MoO_3_ (5 nm) was deposited using a thermal evaporator, and it was sulfurized to the MoS_2_ layer using CVD. The MoO_3_-deposited device was positioned on a thermal furnace under a rotary pump in a CVD chamber. The furnace was heated to ~200 °C, and 3000 slpm hydrogen (H_2_) gas and 20 sccm helium (He) gas was mixed with hydrogen sulfide (H_2_S), 0.1 cmol per mol. Then, very high-frequency (13.56 MHz, 400 W) plasma was applied. Finally, 70 nm of Ag TE was deposited using a thermal evaporator.

Electrical characteristic measurement was conducted under a Probe System using a semiconductor parameter analyzer (Keithley, 4200-SCS).

## Figures and Tables

**Figure 1 materials-14-06275-f001:**
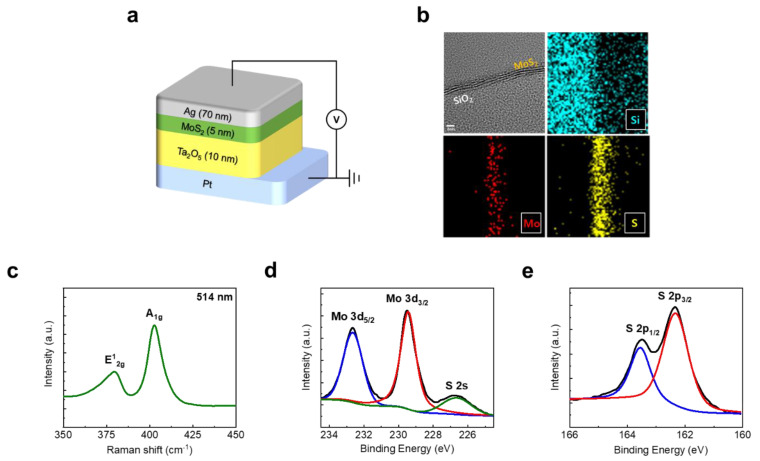
(**a**) Schematic of device structured Pt/Ta_2_O_5_/MoS_2_/Ag. Pt (BE) was grounded and Ag (TE) was applied the voltage. (**b**) The TEM image (top left) and TEM-EDS element mapping image of MoS_2_ film on SiO_2_. (**c**) Raman spectrum of in situ grown MoS_2_ film. (**d**,**e**) XPS spectrum of MoS_2_ film and Mo 3D peaks and S 2p peaks, respectively.

**Figure 2 materials-14-06275-f002:**
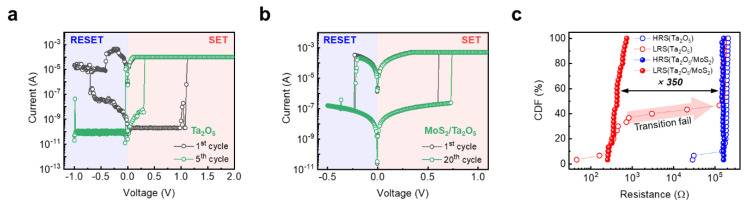
Direct current (DC) current–voltage (I–V) characteristics of Pt/Ta_2_O_5_/Ag device (**a**) and Pt/Ta_2_O_5_/MoS_2_/Ag device (**b**) where the switching cycle of set and reset are up to 5 times and 20 times, respectively. (**c**) Cumulative distributions function (CDF) of LRS and HRS on the Pt/Ta_2_O_5_/Ag device and Pt/Ta_2_O_5_/MoS_2_/Ag device. In the Pt/Ta_2_O_5_/Ag device, the LRS collapsed, but the Pt/Ta_2_O_5_/MoS_2_/Ag device has excellent uniformity, and the average on/off ratio is about 350.

**Figure 3 materials-14-06275-f003:**
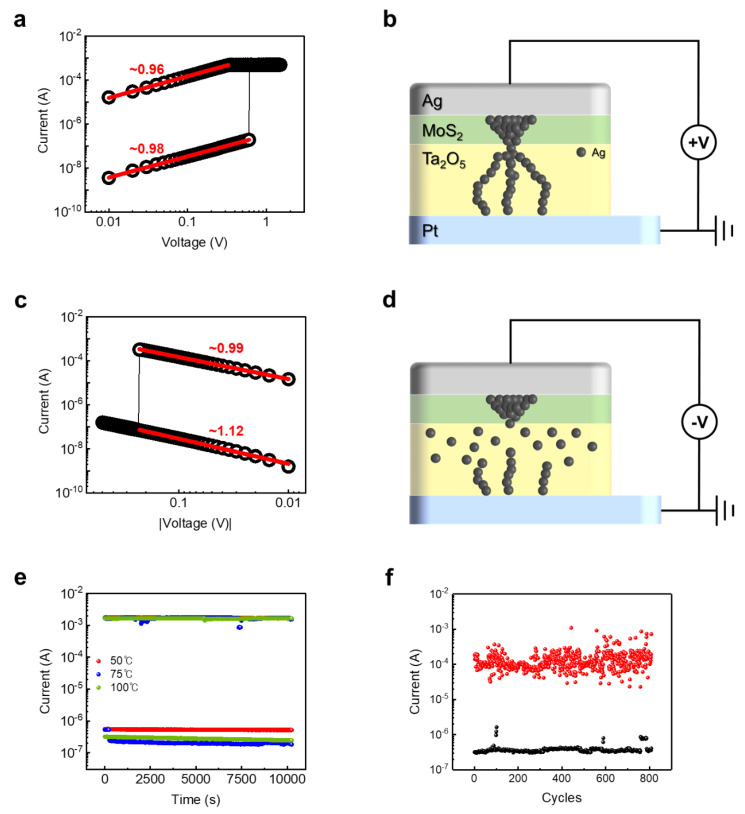
Conduction mechanism and stability for memory application of Pt/Ta_2_O_5_/MoS_2_/Ag device. (**a**,**c**) Double logarithmic I–V characteristics of set and reset processes, respectively, where red solid lines indicate the linear fitting. (**b**,**d**) Schematic illustration of RS mechanism for set and reset processes, respectively. (**e**) Retention properties at 50 °C, 75 °C, and 100 °C for 10,000 s. (**f**) Endurance properties over 800 cycles.

**Figure 4 materials-14-06275-f004:**
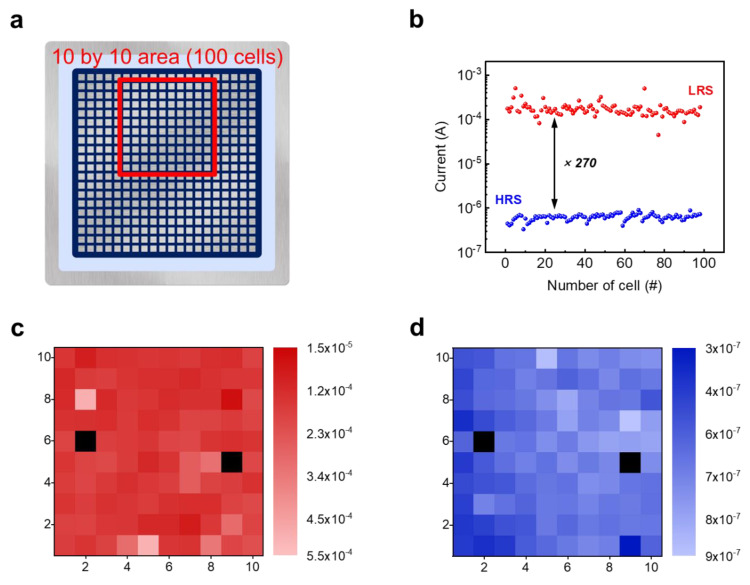
(**a**) Schematic diagram of the device with 400 cells which is structured 20 by 20 matrix, where the area surrounded by red solid line was selected for measurement and the area is 10 by 10 (100 cells). (**b**) Set and reset properties in selected area where the electrical shorted cells were removed. (**c**,**d**) Current mapping image in selected area for set and reset processes, respectively. The black colored squares indicate electrical shorted cells.

**Figure 5 materials-14-06275-f005:**
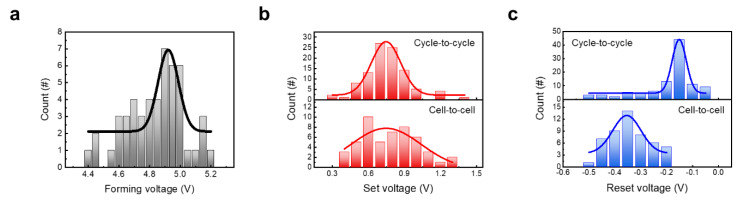
Switching voltage variability. (**a**) Forming voltage variability in 50 cells. (**b**,**c**) Set and reset voltage, respectively. To measure cycle-to cycle and cell-to-cell variability, the switching voltage was extracted in 100 cycles and 50 cells, respectively.

## Data Availability

All data has been included in the paper.

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
