# Peer review of "Statistical Analysis of Uniform Switching Characteristics of Ta2O5-Based Memristors by Embedding In-Situ Grown 2D-MoS2 Buffer Layers"

_materials, 2021, doi:10.3390/ma14216275_

Round 1
Reviewer 1 Report
The manuscript concerns a study about the uniformity of switching characteristics of memristors based on Ta2O5 and demonstrates that using a MoS2 buffer layer improves the uniformity significantly in electrical characteristics in an array of different devices.
In my opinion, the topic is interesting and appropriate for the scope of the journal. The manuscript provides a significant advancement in the field of memristors. I found some minor issues that need clarification to improve the readability and the impact of the article. Once addressed the issues listed below, I expect the manuscript to be suitable for publication in Materials.
Remarks
The authors mention some advantages of MIM configuration in the introduction, as used in the present work. I agree; however, they miss some important general properties that enhance the potentialities of using memristors in RRAM, such as their versatility, the large number of existing systems (see for example Nat. Nanotech. 2013, 8, 13-24), the possible regenerability of the junction (see for example Adv. Mater. 2012, 24, 1197-1201) and so on. This literature, or similar, should be mentioned in the introduction.
The authors stress the properties of memristors based on 2D materials (MoS2); however, the MoS2/Ta2O5 based system cannot be considered a real 2D system. The authors should reconsider the introduction mentioning (stressing) the specific advantages of the Ta2O5 based system.
The authors fabricated a 20x20 array of memristors but showed the data of a 10x10 array. Why?
Why do they select the matrix highlighted in fig 4a? Would you please clarify this point?
I would suggest adding a figure comparing two matrices of devices made with and without the MoS2 buffer. My suggestion is not mandatory, but I believe that It should strongly improve the article's impact.
Author Response
Comments:
The Manuscript concerns a study about the uniformity of switching characteristics of memristors based on Ta2O5 and demonstrates that using a MoS2 buffer layer improves the uniformity significantly in electrical characteristics in an array of different devices.
In my opinion, the topic is interesting and appropriate for the scope of the journal. The manuscript provides a significant advancement in the field of memristors. I found some minor issues that need clarification to improve the readability and the impact of the article. Once addressed the issues listed below, I expect the manuscript to be suitable for publication in Materials.
Reviewer’s comment #1: The authors mention some advantages of MIM configuration in the introduction, as used in the present work. I agree; however, they miss some important general properties that enhance the potentialities of using memristors in RRAM, such as their versatility, the large number of existing systems (see for example Nat. Nanotech. 2013, 8, 13-24), the possible regenerability of the junction (see for example Adv. Mater. 2012, 24, 1197-1201) and so on. This literature, or similar, should be mentioned in the introduction.
Response #1: We appreciated the reviewer’s valuable comment. Following to the reviewer’s comment, we added the two important papers in the number of reference [16, 17] on page 10 as below.
[16] Yang, J. J.; Strukov, D. B.; Stewart, D. R. Memristive Devices For Computing. Nat. Nanotech., 2013, 8, 13–24.
[17] Cavallini, M.; Hemmatian, Z.; Riminucci, A.; Prezioso, M.; Morandi, V.; Murgia, M. Regenerable Resistive Switching in Silicon Oxide Based Nanojunctions. Adv. Mater. 2012, 24, 1197–1201.
Reviewer’s comment #2: The authors fabricated a 20x20 array of memristors but showed the data of a 10x10 array. Why?
Response #2: We appreciated the critical comment. As you know, we already fabricated the 20x20 array of memristor device. However, the edge position cannot be guaranteed in terms of uniformity and film quality. Thus, we selected the 10x10 array region to investigate the device properties. Also, the 100 device cells could be analyzed with high reliability.
Reviewer’s comment #3: Why do they select the matrix highlighted in fig 4a? Would you please clarify this point? I would suggest adding a figure comparing two matrices of devices made with and without the MoS2 buffer. My suggestion is not mandatory, but I believe that It should strongly improve the article's impact.
Response #3: We appreciated the valuable comment. As resembled with comment’s #2, we already explained the reason why we selected the 10x10 array region specifically. Furthermore, we actually intended to compare the device performance between the devices with and without the MoS2 buffer layer as you questioned. However, the operation mechanism of our device is assumed to be filament-type device, meaning that the device reliability is not proper to get the statistical analysis.
We fully tried to answer the comments the reviewers’ suggest and revised the manuscript and Supporting Information. We hope to see get a positive response for the publication on Materials.

Reviewer 2 Report
The authors of manuscript entitled "Statistical Analysis of Uniform Switching Characteristics of Ta2O5-based Memristors by Embedding in-situ Grown 2D-MoS2 Buffer Layers", describe the effect of MoS2 on the performance improvement of Ta2O5-based Memristors. In this work, the author proposes that the multilayer MoS2 (5nm) is used as a buffer layer, which can dramatically stabilize the switching operation compared to pristine device without buffer layer. It is assumed that the difference in mobility and reduction rate of Ta2O5 and MoS2 cause the narrow localization of ion migration, inducing formation more stable conduction filament. However, there is no a direct experimental evidence for the formation and breaking of silver filament, just a conjecture (this is my the biggest concern). I would like to suggest that authors add TEM atomic images in two states (on and off), in particular TEM-EDS elements mapping image. In addition, there are several issues that the authors should clarify.
- There are also some spelling mistakes and improper statements in the manuscript, such as:
- There is an extra period in line 23.
- There is a spelling mistake: spn-transfer in line 46.
- There is a sentence “The experimental data of MoS2 buffered device show the ohmic conduction mechanism in all devices.” in line 77-78. But, the ohmic conduction mechanism is not proved or clarified in the manuscript.
- Grammar mistakes: “Then, the filament started to break to from the relatively weak part.” in line 149.
- The sentence is unclear: “But The Pt/Ta2O5/MoS2/Ag device shows the only non-volatile RS with 350 on/off ratio.” in line 190.
Author Response
Comments:
Response #1: The authors of manuscript entitled "Statistical Analysis of Uniform Switching Characteristics of Ta2O5-based Memristors by Embedding in-situ Grown 2D-MoS2 Buffer Layers", describe the effect of MoS2 on the performance improvement of Ta2O5-based Memristors. In this work, the author proposes that the multilayer MoS2 (5nm) is used as a buffer layer, which can dramatically stabilize the switching operation compared to pristine device without buffer layer. It is assumed that the difference in mobility and reduction rate of Ta2O5 and MoS2 cause the narrow localization of ion migration, inducing formation more stable conduction filament. However, there is no a direct experimental evidence for the formation and breaking of silver filament, just a conjecture (this is my biggest concern). I would like to suggest that authors add TEM atomic images in two states (on and off), in particular TEM-EDS elements mapping image. In addition, there are several issues that the authors should clarify.
Response #1: We appreciated the reviewer’s critical comment. As you already mentioned, it is necessary to clarify the switching operation mechanism by obtaining the cross-TEM atomic images with on and off states, respectively. However, there are some critical huddles to not obtain the TEM image and EDS analysis. First, the actual device area of our memristor has 250 nm via-hole structure as shown in Figure R1. Thus, it is hardly difficult to find the precise position near Ta2O5 switching layer with via-hole structure.
Figure R1. The vertical structure of 250nm via-hole with Pt/Ti bottom electrode on SiO2 substrate.
Second, the in-situ capture analysis in cross-TEM image in needed for special purposed device structure matched with TEM grid with electrical bias condition. However, it is necessary to fabricate the additional device processing. Following to the reviewer’s comment, we could make and confirm in-situ TEM analysis for the next future work definitely.
Reviewer’s comment #2: There are also some spelling mistakes and improper statements in the manuscript, such as; There is an extra period in line 23.
Response #2: We could remove the extra period.
Reviewer’s comment #3: There is a spelling mistake; spn-transfer in line 46.
Response #3: We could fix the from spn-transfer to spin-transfer.
Reviewer’s comment #4: There is a sentence “the experimental data of MoS2 buffered device show the ohmic conduction mechanism in all devices.” In line 77-78. But, the ohmic conduction mechanism is not proved or clarified in the manuscript.
Response #4: We appreciated the reviewer’s valuable comment. In order to investigate the current conduction mechanism, we could perform the analysis by plotting the doule logarithimic plot of I-V curves as shown in Figure R2. For set process as shown in Figure 3a, the resistance states were changed very sharply, resulting that the slope values by fitting the switching curves were extracted around 1. Also, the slope values for reset process were obtained nearly 1. It could be elucidated that the conduction mechanism for 2D-MoS2 buffered devices follows the Ohmic conduction mechanism, which is due to thermally produced free charge carriers. This explanation could be already mentioned in line 123-133 in original manuscript.
Figure R2. The Double logarithmic I-V characteristics of set (a) and reset process (b), respectively, where red solid lines indicate the linear fitting.
Reviewer’s comment #5: Grammar mistake; “Then, the filament started to break to from the relatively weak part.” In line 149.
Response #5: We could revise the sentence as below.
“Then, the filament started to break from the relatively weak part.”
Reviewer’s comment #6: The sentence is unclear: “But Pt/ Pt/Ta2O5/MoS2/Ag device shows the only non-volatile RS with 350 on/off ratio.” in line 190.
Response #6: We appreciated the reviewer’s critical comment. Following to the reviewer’s comment, we could revise more clear sentence as below.
“But, Pt/Ta2O5/MoS2/Ag device shows highly reliable switching performance with minimized device failure with on/off ratio of 350.”
We fully tried to answer the comments the reviewers’ suggest and revised the manuscript and Supporting Information. We hope to see get a positive response for the publication on Materials.

Round 2
Reviewer 2 Report
Authors have almost replied my questions. However, they don’t reply my biggest concern point, TEM atomic images in two states (on and off), in particular TEM-EDS elements mapping image. Authors explain that is difficult question, but they have obtained Fig.1b. Unless Fig.1b obtained by authors is not from the device. In this case, it is not an in situ characteristics and this data cannot be used to describe the thickness and quality of the molybdenum disulfide grown by authors. Although this TEM sample preparation is very difficult, I still recommend the authors supplement this data.
Author Response
We appreciated the reviewer’s valuable comment. Following to the reviewer’s comment, we performed the TEM and EDS mapping analysis on our device with Ta2O5/MoS2/Ag stacks as shown in Figure R1. The left figure in Figure R1 shows the cross-sectional transmission-electron microscopy (TEM) image as a vertical stack with Ta2O5/MoS2/Ag thin film. In order to investigate the EDS mapping in switching operation, the elemental EDS mapping data was shown in right in Figure R1. The Ag element (red line) was obviously detected in the MoS2 film (blue circle region). This means that Ag element in Ag electrode might easily diffuse within grain boundary of 2D-MoS2 film. Also, Ag metal filament movement could be limited to the path of the grain boundary, resulting that the stable filamentary path was formed. As a result, the device switching uniformity was dramatically enhanced.
